# Learning Long-term Visual Dynamics with Region Proposal Interaction Networks

**Haozhi Qi**
UC Berkeley

**Xiaolong Wang**
UC San Diego

**Deepak Pathak**
CMU

**Yi Ma**
UC Berkeley

**Jitendra Malik**
UC Berkeley

## Abstract

Learning long-term dynamics models is the key to understanding physical common sense. Most existing approaches on learning dynamics from visual input sidestep long-term predictions by resorting to rapid re-planning with short-term models. This not only requires such models to be super accurate but also limits them only to tasks where an agent can continuously obtain feedback and take action *at each step* until completion. In this paper, we aim to leverage the ideas from success stories in visual recognition tasks to build object representations that can capture inter-object and object-environment interactions over a long-range. To this end, we propose *Region Proposal Interaction Networks (RPIN)*, which reason about each object's trajectory in a latent region-proposal feature space. Thanks to the simple yet effective object representation, our approach outperforms prior methods by a significant margin both in terms of prediction quality and their ability to plan for downstream tasks, and also generalize well to novel environments. Code, pre-trained models, and more visualization results are available at our Website.

## 1 Introduction

As argued by Kenneth Craik, *if an organism carries a model of external reality and its own possible actions within its head, it is able to react in a much fuller, safer and more competent manner to emergencies which face it* (Craik, 1952). Indeed, building prediction models has been long studied in computer vision and intuitive physics. In computer vision, most approaches make predictions in pixel-space (Denton & Fergus, 2018; Lee et al., 2018; Ebert et al., 2018b; Jayaraman et al., 2019; Walker et al., 2016), which ends up capturing the optical flow (Walker et al., 2016) and is difficult to generalize to long-horizon. In intuitive physics, a common approach is to learn the dynamics directly in an abstracted state space of objects to capture Newtonian physics (Battaglia et al., 2016; Chang et al., 2016; Sanchez-Gonzalez et al., 2020). However, the states end up being detached from raw sensory perception. Unfortunately, these two extremes have barely been connected. In this paper, we argue for a middle-ground to treat images as a window into the world, i.e., objects exist but can only be accessed via images. Images are neither to be used for predicting pixels nor to be isolated from dynamics. We operationalize it by learning to extract a rich state representation directly from images and build dynamics models using the extracted state representations.

<p align="center">It is difficult to make predictions, especially about the future — Niels Bohr</p>

Contrary to Niels Bohr, predictions are, in fact, easy if made only for the short-term. Predictions that are indeed difficult to make and actually matter are the ones made over the long-term. Consider the example of "Three-cushion Billiards" in Figure 1. The goal is to hit the cue ball in such a way that it touches the other two balls and contacts the wall thrice before hitting the last ball. This task is extremely challenging even for human experts because the number of successful trajectories is very sparse. Do players perform classical Newtonian physics calculations to obtain the best action before each shot, or do they just memorize the solution by practicing through exponentially many configurations? Both extremes are not impossible, but often impractical. Players rather build a physical understanding by experience (McCloskey, 1983; Kubricht et al., 2017) and plan by making intuitive, yet accurate predictions in the long-term.

Learning such a long-term prediction model is arguably the "Achilles' heel" of modern machine learning methods. Current approaches on learning physical dynamics of the world *cleverly* side-step the long-term dependency by re-planning at each step via model-predictive control (MPC) (Allgöwer & Zheng, 2012; Camacho & Alba, 2013). The common practice is to train short-term dynamical models (usually 1-step) in a simulator. However, small errors in short-term predictions can accumulate

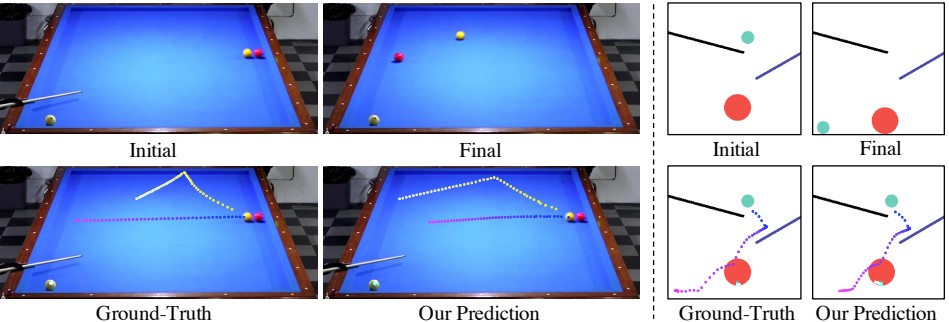

Figure 1: Two example of long-term dynamics prediction tasks. Left: three-cushion billiards. Right: PHYRE intuitive-physics dataset (Bakhtin et al., 2019). Our proposed approach makes accurate long-term predictions that do not necessarily align with the ground truth but provide strong signal for planning.

over time in MPC. Hence, in this work, we focus primarily on the long-term aspect of prediction by just considering environments, such as the three-cushion billiards example or the PHYRE (Bakhtin et al., 2019) in Figure 1, where an agent is allowed to take *only one* action in the beginning so as to preclude any scope of re-planning.

How to learn an accurate dynamics model has been a popular research topic for years. Recently, there are a series of work trying to represent video frames using object-centric representations (Battaglia et al., 2016; Watters et al., 2017; Chang et al., 2016; Janner et al., 2019; Ye et al., 2019; Kipf et al., 2020). However, those methods either operate in the state space, or ignore the environment information, both of which are not practical in real-world scenarios. In contrast, our objective is to build a data-driven prediction model that can both: (a) model long-term interactions over time to plan successfully for new instances, and (b) work from raw visual input in complex real-world environments. Therefore, the question we ask is: how to extract such an effective and flexible object representation and perform long-term predictions?

We propose Region Proposal Interaction Network (RPIN) which contains two key components. Firstly, we leverage the region of interests pooling (RoIPooling) operator (Girshick, 2015) to extract object features maps from the frame-level feature. Object feature extraction based on region proposals has achieved huge success in computer vision (Girshick, 2015; He et al., 2017; Dai et al., 2017; Gkioxari et al., 2019), and yet, surprisingly under-explored in the field of intuitive physics. By using RoIPooling, each object feature contains not only its own information but also the context of the environment. Secondly, we extend the Interaction Network and propose Convolutional Interaction Networks that perform interaction reasoning on the extracted RoI features. Interaction Networks is originally proposed in (Battaglia et al., 2016), where the interaction reasoning is conducted via MLPs. By changing MLPs to convolutions, we can effectively utilize the spatial information of an object and make accurate future prediction of object location and shapes changes.

Notably, our approach is simple, yet outperforms the state-of-the-art methods in both simulation and real datasets. In Section 5, we thoroughly evaluate our approach across four datasets to study scientific questions related to a) prediction quality, b) generalization to time horizons longer than training, c) generalization to unseen configurations, d) planning ability for downstream tasks. Our method reduces the prediction error by 75% in the complex PHYRE environment and achieves state-of-the-art performance on the PHYRE reasoning benchmark.

## 2 RELATED WORK

**Physical Reasoning and Intuitive Physics.** Learning models that can predict the changing dynamics of the scene is the key to building physical common-sense. Such models date back to "NeuroAnimator" (Grzeszczuk et al., 1998) for simulating articulated objects. Several methods in recent years have leveraged deep networks to build data-driven models of intuitive physics (Bhattacharyya et al., 2016; Ehrhardt et al., 2017; Fragkiadaki et al., 2015; Chang et al., 2016; Stewart & Ermon, 2017). However, these methods either require access to the underlying ground-truth state-space or do not scale to long-range due to absence of interaction reasoning. A more generic

yet explicit approach has been to leverage graph neural networks (Scarselli et al., 2009) to capture interactions between entities in a scene (Battaglia et al., 2018; Chang et al., 2016). Closest to our approach are interaction models that scale to pixels and reason about object interaction (Watters et al., 2017; Ye et al., 2019). However, these approaches either reason about object crops with no context around or can only deal with a predetermined number and order of objects. A concurrent work (Girdhar et al., 2020) studies using prediction for physical reasoning, but their prediction model is either in the state space or in the pixel space.

Other common ways to measure physical understanding are to predict future judgments given a scene image, e.g., predicting the stability of a configuration (Groth et al., 2018; Jia et al., 2015; Lerer et al., 2016; Li et al., 2016a;b). Several hybrid methods take a data-driven approach to estimate Newtonian parameters from raw images (Brubaker et al., 2009; Wu et al., 2016; Bhat et al., 2002; Wu et al., 2015), or model Newtonian physics via latent variable to predict motion trajectory in images (Mottaghi et al., 2016a;b; Ye et al., 2018). An extreme example is to use an actual simulator to do inference over objects (Hamrick et al., 2011). The reliance on explicit Newtonian physics makes them infeasible on real-world data and un-instrumented settings. In contrast, we take into account the context around each object via RoIPooling and explicitly model their interaction with each other or with the environment without relying on Newtonian physics, and hence, easily scalable to real videos for long-range predictions.

**Video Prediction.** Instead of modeling physics from raw images, an alternative is to treat visual reasoning as an image translation problem. This approach has been adopted in the line of work that falls under video prediction. The most common theme is to leverage latent-variable models for predicting future (Lee et al., 2018; Denton & Fergus, 2018; Babaeizadeh et al., 2017). Predicting pixels is difficult so several methods leverage auxiliary information like back/fore-ground (Villegas et al., 2017a; Tulyakov et al., 2017; Vondrick et al., 2016), optical flow (Walker et al., 2016; Liu et al., 2017), appearance transformation (Jia et al., 2016; Finn et al., 2016; Chen et al., 2017; Xue et al., 2016), etc. These inductive biases help in a short interval but do not capture long-range behavior as needed in several scenarios, like playing billiards, due to lack of explicit reasoning. Some approaches can scale to relative longer term but are domain-specific, e.g., pre-defined human-pose space (Villegas et al., 2017b; Walker et al., 2017). However, our goal is to model long-term interactions not only for prediction but also to facilitate planning for downstream tasks.

**Learning Dynamics Models.** Unlike video prediction, dynamics models take actions into account for predicting the future, also known as forward models (Jordan & Rumelhart, 1992). Learning these forward dynamics models from images has recently become popular in robotics for both specific tasks (Wahlström et al., 2015; Agrawal et al., 2016; Oh et al., 2015; Finn et al., 2016) and exploration (Pathak et al., 2017; Burda et al., 2019). In contrast to these methods where a deep network directly predicts the whole outcome, we leverage our proposed region-proposal interaction module to capture each object trajectories explicitly to learn long-range forward dynamics as well as video prediction models.

**Planning via Learned Models.** Leveraging models to plan is the standard approach in control for obtaining task-specific behavior. Common approach is to re-plan after each action via Model Predictive Control (Allgöwer & Zheng, 2012; Camacho & Alba, 2013; Deisenroth & Rasmussen, 2011). Scaling the models and planning in a high dimensional space is a challenging problem. With deep learning, several approaches shown promising results on real-world robotic tasks (Finn et al., 2016; Finn & Levine, 2017; Agrawal et al., 2016; Pathak et al., 2018). However, the horizon of these approaches is still very short, and replanning in long-term drifts away in practice. Some methods try to alleviate this issue via object modeling (Janner et al., 2019; Li et al., 2019) or skip connections (Ebert et al., 2018a) but assume the models are trained with state-action pairs. In contrast to prior works where a short-range dynamic model is unrolled in time, we learn our long-range models from passive data and then couple them with short-range forward models to infer actions during planning.

## 3 REGION PROPOSAL INTERACTION NETWORKS

Our model takes $N$ video frames and the corresponding object bounding boxes as inputs, and outputs the objects' bounding boxes and masks for the future $T$ timesteps. The overall model structure is illustrated in Figure 2.

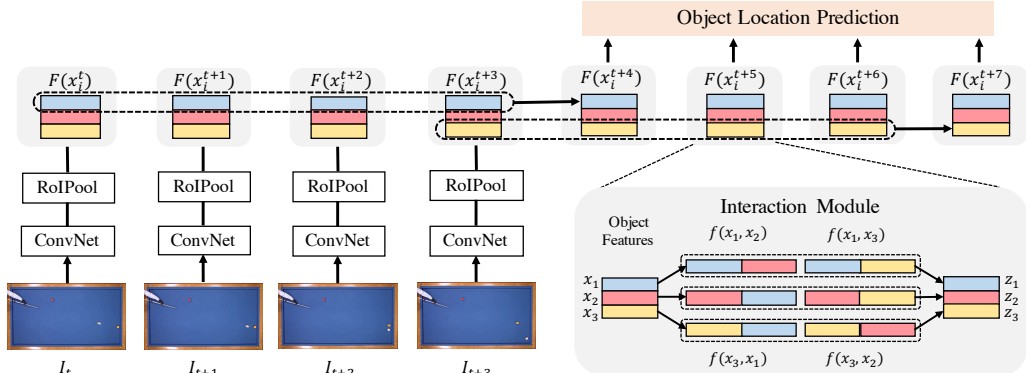

Figure 2: Our Region Proposal Interaction Network. Given $N$ frames as inputs, we forward them to an encoder network, and then extract the foreground object features with RoIPooling (different colors represent different instances). We then perform interaction reasoning on top of the region proposal features (gray box on the bottom right). We predict each future object feature based on the previous $k$ time steps. We then estimate the object location from each object feature.

For each frame, we first extract the image features using a ConvNet. Then we apply RoIPooling (Girshick, 2015; He et al., 2017) to obtain the object-centric visual features. These features are then forwarded to our Convolutional Interaction Networks (CIN) to perform objects' interaction reasoning and used to predict future object bounding boxes and masks. The whole pipeline is trained end-to-end by minimizing the loss between the predicted outputs and the ground-truth.

## 3.1 OBJECT-CENTRIC REPRESENTATION

We apply the houglass network (Newell et al., 2016) to extract the image features. Given an input RGB image $I \in \mathbb{R}^{3 \times H \times W}$, the hourglass network firstly downsample the input via one 2-strided convolution and 2-strided max pooling layers, and then refine the representation by a U-Net-like encoder-decoder modules (Ronneberger et al., 2015). The hourglass network provides features with a large receptive field and a fine spatial resolution (4 times smaller than the input image), both of which are crucial to accurately model object-object and object-environment interactions. We denote the number of output channels of the feature map as $d$.

On top of this feature map, we use RoIPooling (Girshick, 2015) to extract a $d \times h \times w$ object-centric features. RoIPooling takes the feature map and a object bounding box as input. The region corresponding to the bounding box on the feature map is cropped and resize to a fixed spatial size (denoted as $h \times w$). We use $x_i^t$ to represent the feature at $t$-th timestep for the $i$-th object. Such feature representation differs from previous method in two perspectives: 1) It is extracted from image features with a large receptive field, which gives plenty of context information around the object. 2) The feature representation is a 3-dimensional feature map rather than a single vector representation. It can represent the objects' shapes while the vector representation cannot because the spatial dimension is flattened.

## 3.2 CONVOLUTIONAL INTERACTION NETWORKS

To better utilize the spatial information of our object feature map, we propose Convolutional Interaction Networks, an extension of Interaction Network operators on 3-dimensional tensors. In this section, we first briefly review the original Interaction Network (IN) (Battaglia et al., 2016; Watters et al., 2017) and introduce the proposed Convolutional Interaction Networks (CIN).

The original interaction network is a general-purpose data-driven method to model and predict physical dynamics. It takes the feature vectors of $m$ objects at timestep $t$: $X = \{x_1^t, x_2^t, ..., x_m^t \mid x_i^t \in \mathbb{R}^d\}$ and performs object reasoning $f_O$ as well as relational reasoning $f_R$ on these features. Specifically, the updated rule of object features can be described as:

$$
\begin{aligned}
e_i^t &= f_A\big(f_O(x_i^t) + \sum_{j \neq i} f_R(x_i^t, x_j^t)\big), \\
z_i^t &= f_Z(x_i^t, e_i^t), \\
x_i^{t+1} &= f_P\big(z_i^t, z_i^{t-1}, \ldots, z_i^{t-k}\big).
\end{aligned}
\tag{1}
$$

In the above equation, $f_A$ is the function to calculate the effect of both of object reasoning and relational reasoning results. And $f_Z$ is used to combine the original object state and the reasoning effect. Finally, $f_P$ is used to do future state predictions based on one or more previous object states. In IN, $f_{O,R,A,Z,P}$ are instantiated by a fully-connected layer with learnable weights.

**Convolutional Interaction Networks.** The input of CIN is $m$ object feature maps at timestep $t$: $X = \{x_1^t, x_2^t, ..., x_m^t | x_i^t \in \mathbb{R}^{d \times h \times w}\}$. The high-level update rule is the same as IN, but the key difference is that we use convolution to instantiate $f_{O,R,A,Z,P}$. Such instantiation is crucial to utilize the spatial information encoded in our object feature map and to effectively reason future object states. Specifically, we have

$$f_R(x_i^t, x_j^t) = W_R * [x_i^t, x_j^t] \qquad f_O(x_i^t) = W_O * x_i^t$$
$$f_A(x_i^t) = W_A^T * x_i^t \qquad f_Z(x_i^t, e_i^t) = W_Z * [x_i^t, e_i^t] \tag{2}$$

$$f_P(z_i^t, z_i^{t-1}, ..., z_i^{t-k}) = W_P * [z_i^t, z_i^{t-1}, ..., z_i^{t-k}] \tag{3}$$

One can plug the functions in Equation 1 for better understanding the operations. In the above equations, $*$ denotes the convolution operator, and $[\cdot, \cdot]$ denotes concatenation along the channel dimension. $W_R, W_Z \in \mathbb{R}^{d \times 2d \times 3 \times 3}$, $W_O, W_A \in \mathbb{R}^{d \times d \times 3 \times 3}$, $W_P \in \mathbb{R}^{d \times (kd) \times 3 \times 3}$ are learnable weights of the convolution kernels with kernel size $3 \times 3$. We also add ReLU activation after each of the convolutional operators.

### 3.3 LEARNING REGION PROPOSAL INTERACTION NETWORKS (RPIN)

Our model predicts the future bounding box and (optionally) masks of each object. Given the predicted feature $x_i^{t+1}$, we use a simple two layer MLP decoder to estimate its bounding boxes coordinates and masks. The bounding box decoder takes a flattened object feature map as input. It firstly projects it to $d$ dimensional vector, and outputs 4-d vector, representing the center location and size of the box. The mask decoder is of the same architecture but has $21 \times 21$ output channels, representing a $21 \times 21$ binary masks inside the corresponding bounding boxes. We use the $\ell^2$ loss for bounding box predictions. For mask prediction, we use spatial cross-entropy loss which sums the cross-entropy values of a $21 \times 21$ predicted positions. The objective can be written as:

$$L_p = \sum_{t=1}^{T} \lambda_t \sum_{i=1}^{n} \left( \|\hat{B}_i^{t+1} - B_i^{t+1}\|_2^2 + CE(\hat{M}_i^{t+1}, M_i^{t+1}) \right). \tag{4}$$

We use discounted loss during training (Watters et al., 2017) to mitigate the effect of inaccurate prediction at early training stage and $\lambda_t$ is the discounted factor.

## 4 EXPERIMENTAL SETUP

**Datasets.** We evaluate our method's prediction performance on four different datasets, and demonstrate the ability to perform downstream physical reasoning and planning tasks on two of them. We briefly introduce the four datasets below. The full dataset details are in the appendix.

*PHYRE:* We use the BALL-tier of the PHYRE benchmark (Bakhtin et al., 2019). In this dataset, we set $T = 5$. We treat all of the moving balls or jars as objects and other static bodies as background. The benchmark provides two evaluation settings: 1) within task generalization (PHYRE-W), where the testing environments contain the same object category but different sizes and positions; 2) cross task generalization (PHYRE-C), where environments containing objects and context never present during training. We report prediction using the official fold 0 and the physical reasoning performance averaged on 10 folds.

*ShapeStacks (SS)*: This dataset contains multiple stacked objects (cubes, cylinders, or balls) (Ye et al., 2019). In this dataset, we set $T = 15$. We evaluate all baselines and our methods follow the protocol of (Ye et al., 2019) with uncertainty estimation incorporated (see Appendix for detail).

*Real World Billiards (RealB)*: We collect "Three-cushion Billiards" videos from professional games with different viewpoints downloaded from YouTube. There are 62 training videos with $18,306$ frames, and 5 testing videos with $1,995$ frames. The bounding box annotations are from an off-the-shelf ResNet-101 FPN detector (Lin et al., 2017) pretrained on COCO (Lin et al., 2014) and

fine-tuned on a subset of 30 images from our dataset. We manually filtered out wrong detections. In this dataset, we set $T = 20$.

*Simulation Billiards (SimB)*: We create a simulated billiard environment with three different colored balls with a radius 2 are randomly placed in a $64 \times 64$ image. At starting point, one ball is moving with a randomly sampled velocity. We generate 1,000 video sequences for training and 1,000 video sequences for testing, with 100 frames per sequence. We will also evaluate the ability to generalize to more balls and different sized balls in the experiment section. In this dataset, we set $T = 20$.

For PHYRE and SS, it is possible to infer the future from just the initial configuration. So we set $N = 1$ in these two datasets. For SimB and RealB, the objects are moving in a flat table so we cannot infer the future trajectories based on a single image. Therefore in this setting, we set $N = 4$ to infer the velocity/acceleration of objects. We set $k = N$ in all the dataset because we only have access to $N$ features when we make prediction at $t = N + 1$. We predict object masks for PHYRE and ShapeStacks, and only predict bounding boxes for Billiard datasets.

**Baselines.** There are a large amount of work studying how to estimate objects' states and predict their dynamics from raw image inputs. They fall into the following categories:

*VIN:* Instead of using object-centric spatial pooling to extract object features, it use a ConvNet to globally encode an image to a fixed $d \times m$ dimensional vector. Different channels of the vector is assigned to different objects (Kipf et al., 2020; Watters et al., 2017). This approach requires specifying a fixed number of objects and a fixed mapping between feature channels and object identity, which make it impossible to generalize to different number of objects and different appearances.

*Object Masking (OM):* This approach takes one image and $m$ object bounding boxes or masks as input (Wu et al., 2017; Veerapaneni et al., 2019; Janner et al., 2019). For each proposal, only the pixels inside object proposals are kept while others are set to $0$, leading to $m$ masked images. This approach assumes no background information is needed thus fails to predict accurate trajectories in complex environments such as PHYRE. And it also cost $m$ times computational resources.

*CVP:* The object feature is extracted by cropping the object image patch and forwarding it to an encoder (Ye et al., 2019; Yi et al., 2020). Since the object features are directly extracted from the raw image patches, the context information is also ignored. We re-implement CVP's feature extraction method within our framework. We show we can reproduce their results in Section A.2.

## 5 Evaluation Results: Prediction, Generalization, and Planning

We organize this section and analyze our results by discussing four scientific questions related to the prediction quality, generalization ability across time & environment configurations, and the ability to plan actions for downstream tasks.

### 5.1 How accurate is the predicted dynamics?

To evaluate how well the world dynamics is modeled, we first report the average prediction errors on the test split, over *the same time-horizon* as which model is trained on, i.e., $t \in [0, T_{\text{train}}]$. The prediction error is calculated by the squared $\ell_2$ distance between predicted object center location and the ground-truth object centers. The results are shown in Table 1 (left half).

Firstly, we show the effectiveness of our proposed RoI Feature by comparing Table 1 VIN, OM, CVP, and Ours (IN). These four entries use the same backbone network and interaction network modules and only visual encoder is changed. Among them, VIN cannot even be trained on PHYRE-W since it cannot handle varying number of objects. The OM method performs slightly better than other baselines since it also explicitly models objects by instance masking. For CVP, it cannot produce reasonable results on all of the datasets except on the SS dataset. The reason is that in PHYRE and billiard environments, cropped images are unaware of environment information and the relative position and pose of different objects, making it impossible to make accurate predictions. In SS, since the object size is large and objects are very close to each other, the cropped image regions already provide enough context. In contrast, our RoI Feature can implicitly encode the context and environment information and it performs much better than all baselines. In the very challenging PHYRE dataset, the prediction error is only $1/4$ of the best baseline. In the other three easier datasets, the gap is not as large since the environment is less complicated, but our method still achieves

| method | visual encoder | $t \in [0, T_{\text{train}}]$ | | | | $t \in [T_{\text{train}}, 2 \times T_{\text{train}}]$ | | | |
| --- | --- | --- | --- | --- | --- | --- | --- | --- | --- |
| | | PHYRE-W | SS | RealB | SimB | PHYRE-W | SS | RealB | SimB |
| VIN | Global Encoding | N.A. | 2.47 | 1.02 | 3.89 | N.A. | 7.77 | 5.11 | 29.51 |
| OM | Masked Image | 6.45 | 3.01 | 0.59 | 3.48 | 25.72 | 9.51 | 3.23 | 28.87 |
| CVP | Cropped Image | 60.12 | 2.84 | 3.57 | 80.01 | 79.11 | 7.72 | 6.63 | 108.56 |
| Ours (IN) | RoI Feature | 1.50 | 1.85 | 0.37 | 3.01 | 12.45 | 4.89 | 2.72 | 27.88 |
| Ours (CIN) | RoI Feature | **1.31** | **1.03** | **0.30** | **2.55** | **11.10** | **4.73** | **2.34** | **25.77** |

Table 1: We compare our method with different baselines on all four datasets. The left part shows the prediction error when rollout timesteps is the same as training time. The right part shows the generalization ability to longer horizon unseen during training. The error is scaled by 1,000. Our method has significantly improvements on all of the datasets

| method | visual encoder | PHYRE-C | SS-4 | SimB-5 |
| --- | --- | --- | --- | --- |
| VIN | Global Encoding | N.A. | N.A. | N.A. |
| OM | Masked Image | 50.28 | 17.02 | 59.70 |
| CVP | Cropped Image | 99.26 | 16.88 | 113.39 |
| Ours (IN) | RoI Feature | 10.98 | 15.02 | 24.42 |
| Ours (CIN) | RoI Feature | **9.22** | **14.61** | **22.38** |

Table 2: The ability to generalize to novel environments. We show the average prediction error for $t \in [0, 2 \times T_{\text{train}}]$. Our method achieves significantly better results compared to previous methods. The error is scaled by 1,000. Our method generalizes much better than other baselines.

more than 10% improvements. These results clearly demonstrates the advantage of using rich state representations.

Secondly, we show that the effectiveness of our proposed Convolutional Interaction Network by comparing Table 1 Ours (IN) and Ours (CIN). With every other components the same, changing the vector-form representation to spatial feature maps and use convolution to model the interactions can further improve the performance by 10%~40%. This result shows our convolutional interaction network could better utilize the spatial information encoded in the object feature map.

## 5.2 DOES LEARNED MODEL GENERALIZE TO LONGER HORIZON THAN TRAINING?

Generalize to longer horizons is a challenging task. In this section we compare the performance of predicting trajectories longer than training time. In Table 1 (right half), we report the prediction error for $t \in [T_{\text{train}}, 2 \times T_{\text{train}}]$. The results in this setting are consistent with what we found in Section 5.1. Our method still achieves the best performance against all baselines. Specifically, for all datasets except SimB, we reduce the error by more than 30% percent. In SimB, the improvement is not as significant because the interaction with environment only includes bouncing off the boundaries (see Figure 4 (c) and (d)). Meanwhile, changing IN to CIN further improve the performance. This again validates our hypothesis that the key to making accurate long-term feature prediction is the rich state representation extracted from an image.

## 5.3 DOES LEARNED MODEL GENERALIZE TO UNSEEN CONFIGURATIONS?

The general applicability of RoI Feature has been extensively verified in the computer vision community. As one of the benefits, our method can generalize to novel environments configurations without any modifications or online training. We test such a claim by testing on several novel environments unseen during training. Specifically, we construct 1) simulation billiard dataset contains 5 balls with radius 2 (SimB-5); 2) PHYRE-C where the test tasks are not seen during training; 3) ShapeStacks with 4 stacked blocks (SS-4). The results are shown in Table 2.

Since VIN needs a fixed number of objects as input, it cannot generalize to a different number of objects, thus we don't report its performance on SimB-5, PHYRE-C, and SS-4. In the SimB-5 and PHYRE-C setting, where generalization ability to different numbers and different appearances is required, our method reduce the prediction error by 75%. In SS-4, the improvement is not as

|        | Within            | Cross              |
| ------ | ----------------- | ------------------ |
| RAND   | $13.7_{\pm0.5}$   | $13.0_{\pm5.0}$    |
| DQN    | $77.6_{\pm1.1}$   | $36.8_{\pm9.7}$    |
| Ours   | $\mathbf{85.2}_{\pm0.7}$ | $\mathbf{42.2}_{\pm7.1}$ |

Table 3: PHYRE Planning results. RAND stands for a score function with random policy. We show that our method achieves state-of-the-art on both within-task generalization as well as cross-task generalization.

|        | Target State Error | Hitting Accuracy |
| ------ | ------------------ | ---------------- |
| RAND   | 36.91              | 9.50%            |
| CVP    | 29.84              | 20.3%            |
| VIN    | 9.11               | 51.2%            |
| OM     | 8.75               | 54.5%            |
| Ours   | **7.62**           | **57.2%**        |

Table 4: Simulation Billiards planning results. To make a fair comparison, all of baselines and our methods are using the original interaction network and only the visual encoding method is changing. RAND stands for a policy taking random actions.

significant as the previous two because cropped image may be enough on this simpler dataset as mentioned above.

## 5.4 HOW WELL CAN THE LEARNED MODEL BE USED FOR PLANNING ACTIONS?

The advantage of using a general purpose task-agnostic prediction model is that it can help us do downstream planning and reasoning tasks. In this section, we evaluate our prediction model in the recently proposed challenging PHYRE benchmark (Bakhtin et al., 2019) and simulation billiards planning tasks.

**PHYRE.** We use the B-tier of the environment. In this task, we need to place one red ball at a certain location such that the green ball touches another blue/purple object (see figure 1 right for an example). Bakhtin et al. (2019) trains a classification network whose inputs are the first image and a candidate action, and outputs whether the action leads to success. Such a method does not utilize the dynamics information. In contrast, we can train a classifier on top of the predicted objects' features so that it can utilize dynamics information and makes more accurate classification.

During training, we use the same prediction model as described in previous sections except for $T_{\mathrm{train}} = 10$, and then attach an extra classifier on the objects' features. Specifically, we concatenate each object's features at the 0th, 3rd, 6th, and 9th timestep and then pass it through two fully-connected layers to get a feature trajectory for each object. Then we take the average of all the objects' features and pass it through one fully-connected layer to get a score indicating whether this placement solve the task. We minimize the cross-entropy loss between the score and the ground-truth label indicating whether the action is a success. Note that different from previous work (Bakhtin et al., 2019; Girdhar et al., 2020), our model does not need to convert the input image to the 7-channel segmentation map since object information is already utilized by our object-centric representation. During testing, We enumerate the first 10,000 actions from the pre-computed action set in Bakhtin et al. (2019) and render the red ball on the initial image as our prediction model's input. Our final output is an sorted action list according to the model's output score.

We report the AUCCESS metric on the official 10 folds of train/test splits for both within-task generalization and cross-task generalization setting. The results are in Table 3. Our method achieves about 8 points improvement over the strong DQN baseline (Bakhtin et al., 2019) in the within-task generalization setting. On the more challenging setting of cross-task generalization where the environments may not be seen during training, our method is 6 points higher than DQN.

**SimB Planning.** We consider two tasks in the SimB environment: 1) *Billiard Target State.* Given an initial and final configuration after 40 timesteps, the goal is to find one action that will lead to the target configuration. We report the smallest distances between the trajectory between timestep 35-45 and the final position. 2) *Billiard Hitting.* Given the initial configurations, the goal is to find an action that can hit the other two balls within 50 timesteps.

We firstly train a forward model taken image and action as input, to predict the first 4 object positions and render it to image. After that the rendered images are passed in our prediction model. We score each action according to the similarity between the generated trajectory and the goal state. Then the action with the highest score is selected. The results are shown in Table 4. Our results achieves best performance on all tasks. The full planning algorithm and implementation details are included in the appendix.

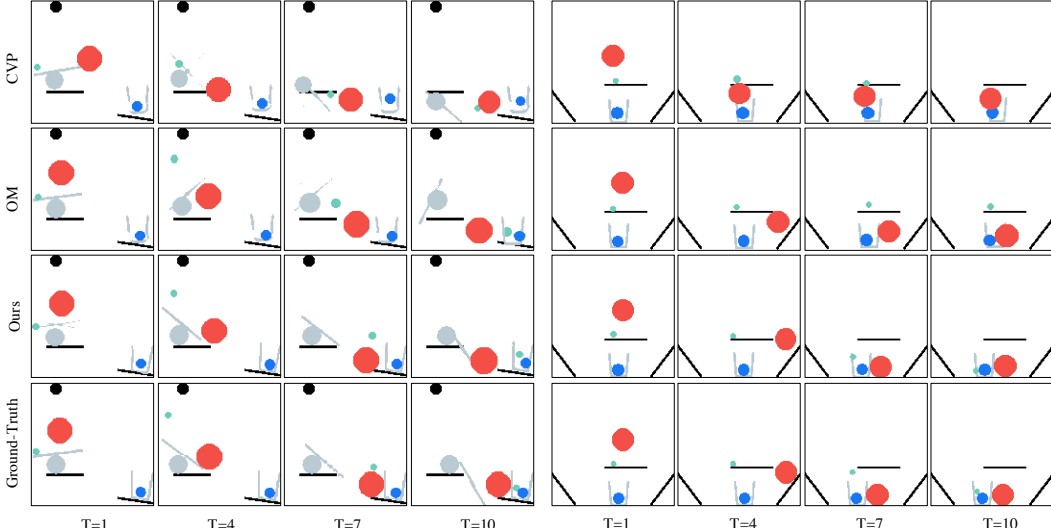

Figure 3: We show the qualitative comparisons between our method and other baselines in the PHYRE benchmark. Our model can accurately infer objects' interactions and collisions with the environment over a long-range (10s), while the baseline methods cannot (the balls sometimes penetrates other objects). In this visualization, only the moving objects' location and masks are predicted (the color is taken from the simulator). Static environment parts (with black color) are rendered from the initial image.

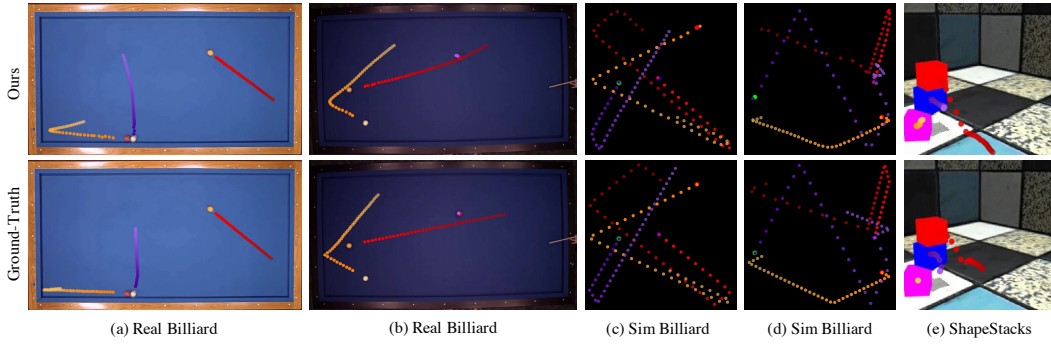

Figure 4: We show the qualitative results of our predicted trajectories in the other three (RealB, SimB, and SS) datasets. Our model could produce accurate and plausible predictions on different scenarios.

## 5.5 QUALITATIVE RESULTS

In figure 3, we show the qualitative prediction results both for our method as well as the OM and CVP baselines. In figure 4, we compare our prediction and ground-truth on the other three datasets. More results and videos are available at our Website.

## 6 CONCLUSIONS

In this paper, we leverage the modern computer vision techniques to propose *Region Proposal Interaction Networks* for physical interaction reasoning with visual inputs. We show that our general, yet simple method achieves a significant improvement and can generalize across both simulation and real-world environments for long-range prediction and planning. We believe this method may serve as a good benchmark for developing future methods in the field of learning intuitive physics, as well as their application to real-world robotics.

## ACKNOWLEDGEMENT

This work is supported in part by DARPA MCS and DARPA LwLL. We would like to thank the members of BAIR for fruitful discussions and comments.

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

# A  IMPLEMENTATION DETAILS

## A.1  NETWORK ARCHITECTURE DETAILS

**Backbone Networks:** We use the same backbone network for our method and all the baselines (VIN, OM, CVP). We choose the hourglass network (Newell et al., 2016) as the image feature extractor. Given an input image, the hourglass network firstly applies a $7\times7$ stride-2 convolution, three residual blocks with channel dimension $d/4$, and a stride-2 max pooling on it. Then this intermediate feature representation is fed into one hourglass modules. In the hourglass module, the feature maps are down-sampled with 3 stride-2 residual blocks and then up-sampled with nearest neighbor interpolation. The dimensions of both the input channel and the output channel of each residual block are $d$. For SimB, we use $d = 64$ since the visual information in this environment is relatively simple. For RealB, PHYRE, and ShapeStacks, $d = 256$. We use batch normalization before each convolutional layer in the backbone network. The resulting feature map size is $d \times \frac{H}{4} \times \frac{W}{4}$, where $H, W$ is the input image size and 4 is the spatial feature stride of the hourglass network.

The output features are transformed to object-centric representations differently for our method and each of the baseline methods:

**VIN:** The resulting feature map is forwarded to 4 convolutional layers with stride 2 and a spatial global average pooling layer to get a $d$ dimensional feature vector. This feature vector is transformed to $d \times m$ by one fully-connected layer. This feature will be reshaped to $m$ vectors with $d$ channel dimensional to represent each object. This representation is refined by two $d \times d$ fully-connected layers and passed to the (convolutional) interaction network.

**OM and CVP:** The output feature map is of shape $m \times d \times \frac{H}{4} \times \frac{W}{4}$ because this method produce $m$ masked (or cropped) images as input. The resulting feature map is forwarded to 4 convolutional layers with stride 2 and a spatial global average pooling layer to get a $d$ dimensional feature vector and the output feature is of size $m \times d$, representing the features of $m$ objects. This representation is refined by two $d \times d$ fully-connected layers and passed to the (convolutional) interaction network.

## A.2  UNCERTAINTY MODELING.

Only in the shapestack datasets, we also incorporate uncertainty estimation follows (Ye et al., 2019), by modeling the latent distribution using a variational auto-encoder (Kingma & Welling, 2014). For the complete details, we refer the reader to (Ye et al., 2019). Here we only give a summary: we build an encoder $h$ which takes the image feature from first $\mathcal{F}^0$ and last frame $\mathcal{F}^T$ of a video sequence as the input. The output of $h$ is a distribution parameter, denoted by $h(\mathbf{u}|\mathcal{F}^0, \mathcal{F}^T)$. Given a particular sample from such distribution, we recover the latent variable by feeding them into a one-layer LSTM and merge into the object feature $x_i^t$. In this case, our pipeline is trained with an additional loss that minimize the KL divergence between the predicted distribution and normal distribution (Kingma & Welling, 2014).

During inference, we sample 100 trajectories for each test image and report the minimum of them. This setting is the same as the original CVP paper Ye et al. (2019). Our implementation gets $5.28 \times 10^{-3}$ squared $\ell_2$ distance for $T \in [0, T_{\text{train}}]$ while the original paper report $6.69 \times 10^{-3}$.

## A.3  DATASET DETAILS

**SimB:** To get the initial velocity, the magnitude (number of pixels moved per timestep) is sampled from $[2, 3, 4, 5, 6]$ and the direction is sampled from $\{6i\pi, i = 0, 1, \ldots, 11\}$.

**RealB:** We found that the bounding box prediction results are accurate enough to serve as the ground-truth. After running the detector, we also manually go through the dataset and filter out images with incorrect detections.

**ShapeStacks:** There are 1,320 training videos and 296 testing videos, with 32 frames per video. Only objects' center positions provided. Following (Ye et al., 2019), we assume the object bounding box is square and of size $70\times70$.

**PHYRE:** For within task generalization (PHYRE-W), the training set contains 80 templates for each of the 25 task. The testing set contains the remaining 20 templates from each task. For cross

task generalization (PHYRE-C), the training set contains 100 templates from 20 tasks while the test set contains 100 templates from the remaining 5 tasks. For each template, we randomly sample a maximum 100 success and 400 failure actions to collect the trajectories to train our model. The image sequence is temporally downsampled by 60.

For our physical reasoning experiments in section 5.4, we train the model using 400 successful actions and 1600 failure actions per template. We use the validation set to select hyper-parameters first, and then do training on the union of train and validation sets and report the final results in the test set.

### A.4 HYPERPARAMETERS

We use Adam optimizer Kingma & Ba (2014) with cosine decay Loshchilov & Hutter (2016) to train our networks. The default input frames is $N = 4$ except $N = 1$ for ShapeStacks and PHYRE. We set $d$ to be 256 except for simulation billiard $d$ is 64. During training, $T$ (denoted as $T_{\text{train}}$) is set to be 20 for SimB and RealB, 5 for PHYRE, and 15 for fair comparison with Ye et al. (2019). The discounted factor $\lambda_t$ is set to be $(\frac{\text{current\_iter}}{\text{max\_iter}})^t$.

*Simulation Billiards.* The image size is $64 \times 64$. We train the model for 100K iterations with a learning rate $2 \times 10^{-3}$, weight decay $1 \times 10^{-6}$, and batch size 200.

*Real World Billiards.* The image is resized to $192 \times 64$. We train the model for 240K iterations with a learning rate $1 \times 10^{-4}$, weight decay $1 \times 10^{-6}$, and batch size 20.

*PHYRE.* The image is resized to $128 \times 128$. We train the model for 150K iterations with a learning rate $2 \times 10^{-4}$, weight decay $3 \times 10^{-7}$, and batch size 20.

*ShapeStacks.* The image is resized to $224 \times 224$. We train the model for 25K iterations with a learning rate $2 \times 10^{-4}$, no weight decay (the same as Ye et al. (2019)), and batch size 40. In this dataset, we apply uncertainty modeling. The loss weight of KL-divergence is $3 \times 10^{-5}$. During inference, following Ye et al. (2019), we randomly sample 100 outputs from our model, and select the best (in terms of the distance to ground-truth) of them as our model's output.

## B PLANNING DETAILS

Given an initial state (represented by an image) and a goal, we aim to produce an action that can lead to the goal from the initial state. Our planning algorithm works in a similar way as visual imagination Fragkiadaki et al. (2015): Firstly, we select a candidate action **a** from a candidate action set $\mathcal{A}$. Then we generate the input images $\mathcal{I}$. For SimB, we train a forward model take the initial image and the action embedding to generate object positions for the next 3 steps. Then we use the simulator to generate the 3 images. For PHYRE, we convert the action to the red ball using the simulator and get the initial image. After that, we run our model described in section 5.4 and the score of each action is simply the classifier's output. We then select the action with the max score.

We introduce the action set for each task in section B.1, and how to design distance function in B.2. A summary of our algorithm is in Algorithm 1.

### B.1 CANDIDATE ACTION SETS

For simulation billiard, the action is 3 dimensional. The first two dimensions stand for the direction of the force. The last dimension stands for the magnitude of the force. During doing planning, we enumerate over 5 different magnitudes and 12 different angles, leading to 60 possible actions. All of the initial condition is guaranteed to have a solution.

For PHYRE, the action is also 3 dimensional. The first two dimensions stand for the location placing the red ball. The last dimension stands for the radius of the ball. Following (Bakhtin et al., 2019), we use the first 10k actions provided by the benchmark.

### B.2 DISTANCE FUNCTION

**Init-End State Error.** Denote the given target location of $m$ objects as $y \in R^{m \times 2}$. We use the following distance function, which measures the distance between the final rollout location and the

target location:

$$D = \sum_{i=1}^{m} \sum_{j=1}^{2} (\hat{p}_{T,i,j} - y_{i,j})^2 \qquad (5)$$

**Hitting Accuracy.** Denote the given initial location of $m$ objects as $x \in R^{m \times 2}$. We apply force at the object $i'$. We use the following distance function, which prefer the larger moving distance for objects other than $i'$:

$$D = -\min_{i} \sum_{i=1, i \neq i'}^{m} \sum_{j=1}^{2} (\hat{p}_{T,i,j} - x_{i,j})^2 \qquad (6)$$

**PHYRE task.** Since we already train a classifier to classify whether the predicted trajectory will lead to a successful solution of the current task, during test time we can directly use it to classify whether the current action will lead to a successful solution. The distance function is just the negative output score.

### B.3 PLANNING ALGORITHM

---
**Algorithm 1:** Planning Algorithm for Simulated Billiard and PHYRE

---
**Input:** candidate actions $\mathcal{A} = \{\mathbf{a}_i\}$, initial state $x$, end state $y$ (optional)
**Output:** action $\mathbf{a}^*$
**for** $\mathbf{a}$ *in* $\mathcal{A}$ **do**
    $\mathcal{I} = \text{Simulation}(x, \mathbf{a})$ ;
    $\hat{p} = \text{PredictionModel}(\mathcal{I})$ ;
    calculate $D$ according to task as in B.2;
    **if** $D < D^*$ **then**
        $D^* = D$ ;
        $a^* = a$ ;
    **end**
**end**

---

## C PHYRE 10 FOLD RESULTS

To enable future work to compare with our method, we provide AUCCESS scores for all folds in Table 5. The number of RAND and DQN is taken from Bakhtin et al. (2019).

| setting | method | fold id | | | | | | | | | |
|---|---|---|---|---|---|---|---|---|---|---|---|
| | | 0 | 1 | 2 | 3 | 4 | 5 | 6 | 7 | 8 | 9 |
| B (within) | RAND | 13.44 | 14.01 | 13.79 | 13.80 | 12.75 | 13.34 | 13.95 | 14.30 | 13.36 | 14.33 |
| | DQN | 76.82 | 79.72 | 78.22 | 75.86 | 77.03 | 78.42 | 78.01 | 77.34 | 78.04 | 76.87 |
| | **Ours** | 85.49 | 86.57 | 85.58 | 84.11 | 85.30 | 85.18 | 84.78 | 84.32 | 85.71 | 85.17 |
| B (cross) | RAND | 11.78 | 12.42 | 18.18 | 12.42 | 3.81 | 22.50 | 11.73 | 13.29 | 8.94 | 14.60 |
| | DQN | 43.69 | 30.96 | 43.05 | 43.91 | 22.77 | 44.40 | 34.53 | 39.20 | 18.98 | 46.46 |
| | **Ours** | 50.86 | 36.58 | 55.44 | 38.34 | 37.11 | 47.23 | 38.23 | 47.19 | 32.23 | 38.76 |

Table 5: The AUCCESS scores for each of evaluated fold.

