# OpenReview forum: "Learning Long-term Visual Dynamics with Region Proposal Interaction Networks"
_ICLR.cc/2021/Conference — ICLR 2021 Poster_

### Official Review · AnonReviewer4 · 2020-10-25
**Promising approach with object centric representations**

**Rating:** 7
**Confidence:** 4

**Review:**

Paper Summary
-----------------------------------
The paper proposes a variation of interaction networks (IN) called region proposal interaction networks. The key idea is to have a richer object-centric feature representation using ROI-Pooling to encode the objects for prediction and use convolution operators to help the IN handle the change in the dimensionality of the feature representation. The paper is well-written and is evaluated on several popular benchmarks. The proposed variations seem to have a considerable effect on the performance to offer significant gains on challenging benchmarks.

Comments:
- The predictions for time t+1 is a function of features from t-k to t. What is the value of 'k' used in the experiments? How is this chosen? Does this change across datasets, environments, tasks, etc.? k has a direct effect on the number of parameters in the network W_p and it would be interesting to see how it affects the predictions.
- On that note, are the object features re-computed using ROI pooling at every time step, or are the predicted features used recursively? Have you tried generating frames based on the predictions to see if it helps capture the interaction/context between the different objects in the scene based on predictions?
- How are the objects registered across time? In other words, is there some tracking used to map the object bounding boxes across time to ensure that consistency is enforced when modeling the interaction of the same object over time? If there is no explicit registration, then is the interaction modeled on a per-frame basis i.e. ground truth is used to ensure that the features of each object are used for its interaction modeling? If the GT is used, then I am not sure how would the model fare without access to such explicit information when encountering unseen objects?
- I am not convinced about the use of the term 'spatial information' when using ROI pooling. From what I can see, ROI pooling results in a 3-d representation of the object as opposed to 1-d representations obtained when using cropped images. This offers a richer representation that captures object-centric features that is invariant to the spatial dimensions of objects. But I do not think that the spatial information is explicitly coded into the features are any point.
- While the paper is well written, there are some inconsistencies in notations that are hard to follow. For example, in Equation 2, the parameters to the function f_Z has 'h' which in the writing refers to the height of the object-centric representation vector. Following consistent notations will greatly improve the readability.
- The paper would benefit from a more thorough ablation study. There are several factors that are used in the proposed framework that could have a considerable effect on the performance but not discussed such as the temporal parameter 'k' mentioned above.

---

> ### Author Response · Authors · 2020-11-19
> **Response to Reviewer #4**
>
> Thank you for your constructive suggestions!
>
> **Q1: “Have you tried generating frames based on the predictions… ?”**
>
> A1: Yes, we had provided a lot of visualizations of rendering from our predictions on the supplementary website: https://sites.google.com/view/iclr21-rpin. The visualization results indicate our model can indeed capture the interaction between multiple objects as well as with the context.
>
>
> **Q2: “What is the value of 'k' used in the experiments? How is this chosen? Does this change across datasets, environments, tasks, etc.? k has a direct effect... and it would be interesting to see how it affects the predictions.”**
>
> A2: k is equal to N in our experiments. The reason is that when we predict the trajectory of t=N+1, we only have access to the previous N features. We revised our submission to explain this point (Section 4, red text).
>
> The value of N depends on the task:
> 1. For PHYRE and ShapeStacks where the objects are governed by gravity, it is possible to infer the future from just the initial configuration. Following (Bakhtin et al., 2019; Ye et al., 2019), we set N=1.
> 2. For SimB and RealB, the objects are moving in a flat table so we cannot infer the future trajectories based on a single image. Therefore in this setting, we set N=4 to infer the velocity/acceleration of objects.
>
> In SimB and RealB, we did experiments with different N (=k), the results do not have significant differences. We choose N=4 due to a balance between performance and computation budgets. The results are in the table below:
>
> |       method      | N |       mean error of [0, $T_{train}$]  | mean error of [$T_{train}$, 2x$T_{train}$]  |
> | ------------------- | --- |  ----------------------------------- | -----------------------------------------------  |
> |                         |     |        RealB        SimB           |          RealB                 SimB |
> |    Ours (CIN)   |  3  |         0.35          2.63            |            2.52                 28.69              |
> |    Ours (CIN)   |  4  |         0.32          2.55            |            2.44                 27.04              |
> |    Ours (CIN)   |  6  |         0.31          2.49            |            2.40                 26.74              |
>
>
> **Q3: “I am not convinced about the use of the term 'spatial information' when using ROI pooling. From what I can see, ROI pooling results in a 3-d representation of the object as opposed to 1-d representations obtained when using cropped images. This offers a richer representation that captures object-centric features that are invariant to the spatial dimensions of objects. But I do not think that the spatial information is explicitly coded into the features are any point.”**
>
> A3: The term 'spatial information' means the objects' shapes which can be implicitly encoded in the RoIPooling features, instead of the x,y spatial locations. We tried to add location features into our framework. While we did observe improvements in less complex environments such as SimB and RealB, it does not help much in more complex environments such as PHYRE (see the table below). We hypothesize this is because location features cannot provide more information to infer the interaction with the environment. We revised our submission to make it clear (Section 3.1, red text).
>
> | Feature              	             	| mean error of [0, $T_{train}$]  | mean error of [$T_{train}$, 2x$T_{train}$]  |
> |-----------------------------------------|-----------------| -----------------|
> |                                      | PHYRE  SS  RealB SimB  |  PHYRE  SS  RealB  SimB |
> | RoI Feature   		   		|  1.70   1.73   0.32   2.55  	| 11.91   4.33   2.44   27.04	|
> | RoI + Position Feature |    1.68   1.70   0.31   2.42   |   11.93   4.28   2.26   26.32 |
>
>
>
> **Q4: “How are objects registered across time? If the GT is used, then I am not sure how would the model fare without access to such explicit information when encountering unseen objects?”**
>
> A4: No, we do not use GT information. For billiards data (RealB and SimB), we perform tracking by the association of object appearance features. For PHYRE and ShapeStacks, we only require 1 image as input, therefore no need to do tracking.
>
>
> **Q5: “Are features re-computed or predicted?”**
>
> A5: Only the first N frames use features from RoIPooling. In later frames, there is no visual input thus we use the predicted features recursively.
>
>
> **Q6: Inconsistency in notation.**
>
> A6: Thank you for your valuable feedback. We revised the paper to fix the notations (Equation 3, red part).
>
>
> Thanks again for the comments. Please let us know if you have any further questions or clarifications.
>
>
> Reference:
>
> Yufei Ye, Maneesh Singh, Abhinav Gupta, and Shubham Tulsiani. Compositional video prediction. ICCV, 2019.
>
> Anton Bakhtin, Laurens van der Maaten, Justin Johnson, Laura Gustafson, and Ross Girshick. Phyre: A new benchmark for physical reasoning. arXiv, 2019.

---

> > ### Comment · AnonReviewer4 · 2020-11-24
> > **Thanks for your response**
> >
> > Thanks for providing the clarifications. I really appreciate your detailed responses. Having read the other reviews and the author's responses, I feel that the paper makes a good contribution in integrating object-centric representations into the prediction process. Overall, I think this is a good paper and would recommend its acceptance. I stand by my original rating.

---

### Official Review · AnonReviewer1 · 2020-10-25
**Leaning towards accept, but low technical novelty**

**Rating:** 6
**Confidence:** 3

**Review:**

*Summary*: The paper presents a DNN for learning dynamical models using the SOTA in deep learning-based computer vision literature.

Strengths:
1. The main strength of the paper is that it uses the SOTA in computer vision (DNNs for image and object understanding) to achieve a different goal in an area in which research has not yet matured (at least to my understanding). By doing so, it helps stimulate and ease further research in the latter area .

2. There are clear experiments that effectively demonstrate the contribution of the paper.

Weaknesses:

1. I would like to have seen some qualitative comparisons with the baselines to get a better sense of the superiority of the method.

2. The technical novelty is low as it seems the main technical contribution is to only replace the MLP from related work with a ConvNet


Comments/Questions to authors:
1. In the introduction, you write *"... those methods operate in the state space, or ignore the environment information"*. Could you elaborate a bit on this point. Folks outside the field may not understand what it means to "operate in the state space" and why it that not good ?

---

> ### Author Response · Authors · 2020-11-19
> **Response to Reviewer #1**
>
> Thank you for your constructive suggestions!
>
> **Q1: “would like to have seen some qualitative comparisons with the baselines to get a better sense of the superiority of the method.”**
>
> A1: Yes, indeed, we had already provided a lot of qualitative comparisons with different baselines in the PHYRE dataset on our supplementary website: https://sites.google.com/view/iclr21-rpin. These visualization results indicate our model can model the bouncing/collision with the environment correctly while the baseline methods cannot do so. For example, please see the example in Row 1 of Section 1, the green object directly penetrates the black/gray bar for the two baseline methods.
>
>
> **Q2: The technical novelty is low.**
>
> A2: We regard the simplicity of our method as an important virtue. Despite being simple, it brings together the key ideas from computer vision (region proposal features) and machine learning (interaction network) together. The advantage of simplicity is further supported by the results where our method significantly outperforms all prior state-of-the-art methods by a healthy margin.
>
>
> **Q3: What is the meaning of ‘operate in the state space’?**
>
> A3: The term ‘state space’ here comes from control theory literature. State here means objects' physical properties such as positions, shapes, sizes, orientation, etc. In the real world, we don't have direct access to the underlying true state, but only sensory observations (e.g., images) to estimate those states. Moreover, even if we can estimate the objects’ states, it is hard to have a general approach to describe the environments using several parameters, such as the billiard table under different viewpoints or the bars/jars in the PHYRE dataset. In contrast, our approach directly takes image observation as input and implicitly encodes the environment information into object features, which is general and practical in the real world. We will clarify it in the paper.
>
>
> Thanks again for the comments. Please let us know if you have any further questions or clarifications.

---

### Official Review · AnonReviewer3 · 2020-10-28
**Simple, effective method to incorporate spatial information in learning visual dynamics**

**Rating:** 7
**Confidence:** 4

**Review:**

The authors propose a novel architecture RPIN (region proposal interacion networks) to learn physical interaction dynamics directly from visual input using region of interest features and a convolutional interaction module which retains spatial information. The authors showcase improvements in 4 different datasets for long term prediction, planning and generalization.


Strengths:
* Simple and effective idea to encode spatial information and use environment information via convolutional and RoI features respectively.
* Improvements on RealB and Phyre-C seem to be very promising (esp in the longer than training horizon setting).


Weaknesses:
* Evaluation and insight into how changing feature representation for interaction networks impacts long term prediction seems to be missing from the paper.

Questions:

1. How does Fp (equation 2) work for scenes with different number of objects as your number of channels would change with changing number of objects in a scene.


2. The authors mention that "In SimB, the improvement is not as significant because there is no environment information needed to infer future dynamics". This doesn't seem to be true as you need scene context to perform prediction even in the simulated environment?

---

> ### Author Response · Authors · 2020-11-19
> **Response to Reviewer #3**
>
> **Q1: “Evaluation and insight into how changing feature representation for interaction networks impact long term prediction seems to be missing from the paper.”**
>
> A1: This is a good suggestion. There are several alternatives one could try to improve the representation for better performance:
>
> 1. Use self-supervised auxiliary loss, e.g. contrastive learning, etc. We believe doing so will help in scenarios when the input scenes are really complex and real-world.
>
> 2. Another way is to augment features with object state information: We tried this idea by augmenting object features using the positional encoding of objects (coordinate of bounding boxes). While we did observe improvements in less complex environments such as SimB and RealB, it does not help much in more complex environments such as PHYRE (see the table below). We hypothesize this is because positional features cannot provide more information to infer the interaction with the environment.
>
> | Feature              	             	| mean error of [0, $T_{train}$]  | mean error of [$T_{train}$, 2x$T_{train}$]  |
> |-----------------------------------------|-----------------| -----------------|
> |                                      | PHYRE  SS  RealB SimB  |  PHYRE  SS  RealB  SimB |
> | RoI Feature   		   		|  1.70   1.73   0.32   2.55  	| 11.91   4.33   2.44   27.04	|
> | RoI + Position Feature |    1.68   1.70   0.31   2.42   |   11.93   4.28   2.26   26.32 |
>
> **Q2: How does F_p work for scenes with a different number of objects?**
>
> A2: Thanks for pointing it out. There was a typo in the Equation of f_p: the index t, t-1, ..., t-k should be superscript instead of the subscript. Please refer to the updated submission file (Equation 3, red part).
> The number of objects only affects the e_i^t in equation (1). The f_p function is applied for each object feature across different timesteps, therefore it is agnostic to the number of objects and can be applied to any number of them.
>
>
> **Q3: “The authors mention that ‘In SimB, the improvement is not as significant because there is no environment information needed to infer future dynamics’. This doesn't seem to be true as you need scene context to perform prediction even in the simulated environment?”**
>
> A3: By ‘interaction with environment’, we mean the collision or bouncing with the environments (such as the bars in the PHYRE dataset or the billiard table in the RealB dataset). In SimB, the environment is less complicated (interaction with environment only includes bouncing off the boundaries), thus our model has less performance gain.
>
> Thanks again for comments. Please let us know if you have any further questions or clarifications.

---

### Official Review · AnonReviewer2 · 2020-10-28
**A tidy model, with impressive results**

**Rating:** 6
**Confidence:** 2

**Review:**

Summary:

This paper introduces a method for predicting future trajectories of sets of objects.
They generate features for each object in the scene using a CNN followed by ROIPooling for each object.
They employ Interaction Networks to implement an update function that uses the relations between the object states.
Because they use spatial features for each object, they update the original Interaction Networks to take spatial blocks as opposed to vectors with convolutions instead of mlp's.
Their models output predicted bounding boxes, but can also be trained to output a mask (21x21) for each predicted object bounding box.

Reason for Score:

I vote for accepting.
The method is clear, and the results are good.
I would be interested in applications of this method to real world video using off the shelf object detectors.

Pros:

ROIPooled object features from CNN outputs have been demonstrated to be packed with information about both the objects and their context.
The enhancement to the original Interaction Networks (using convolutions) seems to bring performance improvements (Table 1).
The videos (especially for PYHRE dataset) are impressive.
The benefits of being able to make long term predictions when planning are clear.

Cons:

I am a little worried about the size of the input images (eg. 128x128 for PYHRE ) and the size of the spatial output of the CNN that is used for ROIPooling of the objects. I cannot see this size in the paper (should be in A.1?). It seems it might be too small to properly use ROIPooling that discriminates between close objects.
I am not sure the baseline comparisons are very fair, since it is stated that they are either video based or do not have mechanism to cope with "background" objects.

---

> ### Author Response · Authors · 2020-11-19
> **Response to Reviewer #2**
>
> Thank you for your constructive suggestions!
>
>
> **Q1: “The size of the spatial output might be too small to discriminate between close objects.”**
>
> A1: The output feature is actually not small. The network has stride 4 and the output is 32x32. Since almost all objects are larger than 4 x 4 in the original image, the feature of each object occupies at least one pixel in the feature map. Thus the network is still able to distinguish close objects. We revised the text to make it more clear in appendix A.1.
>
>
> **Q2: “Baseline results are not a fair comparison since they are either video-based or do not cope with backgrounds.”**
>
> A2: We respectfully disagree with reviewer and believe the comparisons are fair because all the methods do not get any information about boundary or background **explicitly**. However, our method is able to encode the environment information implicitly in its object-centric feature. This is exactly our contribution here: environment information is important, but previous methods can’t capture it. We further show that implicitly modeling the environment by RoIPooling features can achieve much better performance both qualitatively and quantitatively.
>
>
> Thanks again for the comments. Please let us know if you have any further questions or clarifications.

---

### Decision · Program_Chairs · 2021-01-07
**Final Decision**

**Decision:**

Accept (Poster)

**Comment:**

This paper was reviewed by four experts in the field. Based on the reviewers' feedback, the decision is to recommend the paper for acceptance to ICLR 2021. The reviewers did raise some valuable concerns that should be addressed in the final camera-ready version of the paper. The authors are encouraged to make the necessary changes and include the missing references.